# Abnormalities of hippocampus and frontal lobes in heart failure patients and animal models with cognitive impairment or depression: A systematic review

Ziwen Lu[1‡], Yu Teng[1‡], Lei Wang[1], Yangyang Jiang[1], Tong Li[1], Shiqi Chen[1], Baofu Wang[1], Yang Li[1], Jingjing Yang[1], Xiaoxiao Wu[1], Weiting Cheng[1], Xiangning Cui[2]*, Mingjing Zhao[1]*

1 Key Laboratory of Chinese Internal Medicine of Ministry of Education and Dongzhimen Hospital, Beijing University of Chinese Medicine, Beijing, China, 2 Department of Cardiovascular, Guang'anmen Hospital, China Academy of Chinese Medical Sciences, Beijing, China

‡ ZL and YT are share first authorship on this work.
* mjgx2004@163.com (MZ); cuixiangning@126.com (XC)

**Data Availability Statement:** All relevant data are within the paper and its Supporting information files.

## Abstract

### Aims

This systematic review aimed to study the hippocampal and frontal changes of heart failure (HF) patients and HF animal models with cognitive impairment or depression.

### Methods

A systematic review of the literature was conducted independently by reviewers using PubMed, Web of Science, Embase, and the Cochrane Library databases.

### Results and conclusions

30 studies were included, involving 17 pieces of clinical research on HF patients and 13 studies of HF animal models. In HF patients, the hippocampal injuries were shown in the reduction of volume, CBF, glucose metabolism, and gray matter, which were mainly observed in the right hippocampus. The frontal damages were only in reduced gray matter and have no difference between the right and left sides. The included HF animal model studies were generalized and demonstrated the changes in inflammation and apoptosis, synaptic reduction, and neurotransmitter disorders in the hippocampus and frontal lobes. The results of HF animal model studies complemented the clinical observations by providing potential mechanistic explanations of the changes in the hippocampus and frontal lobes.

**Funding:** MJ Z is funded by the National Natural Science Foundation of China (no.81973787). The funders had no role in study design, data collection and analysis, decision to publish, or preparation of the manuscript.

**Competing interests:** The authors have declared that no competing interests exist.

**Abbreviations:** AD, Alzheimer's disease; BAI, the Beck anxiety inventory; BDI-II, the Beck depression inventory; BNP, brain natriuretic peptide; CAMCOG, Cambridge mental disorders of the elderly examination; CBF, cerebral blood flow; CES-D, center for epidemiologic studies depression scale; CI, cognition impairment; CVLT, California verbal learning test; DM, a test of delayed memory; EDA, end-diastolic area; EDD, end-diastolic diameter; EDV, end-diastolic volume; ESA, end-systolic area; ESD, end-systolic diameter; ESV, end-systolic volume; FS, fractional shortening; GDS, Geriatric Depression Scale; GMD, gray matter density; H5PT, hamasch 5-point test revised; HDS-R, the revised Hasegawa's dementia scale; HF, heart failure; HFpEF, heart failure with preserved ejection fraction; HFrEF, heart failure with reduced ejection fraction; IM, immediate memory; LAD, left anterior descending coronary; LVDd, left ventricular diastolic dimension; LVEDD, left ventricular end-diastolic diameter; LVEF, left ventricular ejection fraction; LVESD, left ventricular end systolic diameter; MABP, mean arterial blood pressure; MCI, mild cognitive impairment; MI, myocardial infarction; MMSE, Mini-Mental State Examination; MoCA, Montreal Cognitive Assessment test; NT-proBNP, N-terminal prohormone of brain natriuretic peptide; NYHA, New York Heart Association; PWTd, posterior wall thickness in diastole; RET, the Regensburger Word Fluency Test; ROCF, Rey-Osterrieth complex figure test; RWT, Regensburg word fluency test; STAI, State-Trait Anxiety Inventory; TAP, test battery of attentional processes; TEM, Transmission Electron Microscopy; TMT-A, trail making test A; TMT-B, trail making test B; VFT, verbal fluency task; VVM2, the visual and verbal memory test; WMS-R, the Wechsler Memory Scale revised.

# Introduction

Heart failure (HF) is a rapidly increasingly cardiovascular disease with high morbidity and mortality around the world over the past decade [1]. Numerous researches have investigated that cognition impairment (CI) and depression were prevalent in patients with HF [2, 3], with the incidence of approximately 43–82% and 30%, respectively [3, 4]. HF and cognitive impairment/depression comorbidity increase hospitalization and risk of Alzheimer's disease, impacting the living quality in HF patients [5].

The cognitive declines in HF patients were mild cognitive impairment (MCI) affecting memory, attention, and executive function, which was different from Alzheimer's disease [6]. It has been known that the neurological symptoms were closely bound up with the injury of the brain's specific regions [7, 8]. Hippocampus is related to memory and is also seen as the regulator of emotion [9, 10]. Frontal lobe is critical to acquisition, execution, and control of a wide range of functions, from basic motor response to complex decision-making [11]. The function of the hippocampus and frontal lobe was consistent with cognitive impairment in HF patients, therefore, this review focused on changes in these two regions. Moreover, it has been reported fragmentary studies about imaging of the hippocampus and frontal lobe after HF, however, it lacks systematic summaries and generalizations about damage changes of the hippocampus and frontal lobe after HF. Meanwhile, the animal models could complement the clinical observations by providing potential mechanistic explanations. Therefore, the characteristic changes of the hippocampus and frontal lobe in HF animal model still need to be systematically reviewed.

This systematic review screened the current literature to identify characteristic cognitive impairment and depression in HF. Most importantly, it discussed the changes in the hippocampus and frontal lobe in HF patients, and thus whether these changes had differences between the right and left sides. Then, pathological mechanisms of hippocampal and frontal damage in HF animal model studies were also summarized to complement clinical observations. This systematic review would provide a reference for future clinical prevention and treatment of HF.

# Methods

The report of this systematic review was prepared based on the PRISMA 2020 Statements.

## 1. Information sources and search strategy

The published articles were searched comprehensively in electronic databases (PubMed, Web of Science, Embase, and Cochrane Library) up to August 2022. Keywords and Medical Subject Heading (MeSH) terms used in these searches included heart failure, heart decompensation, myocardial failure, congestive heart failure, cognitive dysfunction, cognitive impairment, cognitive disorder, mental deterioration, depression, depressive symptoms, emotional depression, hippocampus, hippocampus propers, hippocampal formation, ammon horn, subiculum, frontal lobes, frontal cortex, supplementary eye field, and brodmann Area 8. No filter or limitation was used during the search. The detailed search strategy was acquired in supplement materials. For all identified studies, a manual search was conducted of their references and review articles to locate additional relevant studies.

## 2. Inclusion and exclusion criteria

The inclusion criteria of the clinical studies were as follows: (1) Participants: Patients meeting the diagnostic criteria of HF were included, with or without the control group; (2) Method of

research: Cohort studies, case-control studies, cross-sectional studies, two-group comparative studies and prospective studies were included; (3) The study had to involve in the structure and function of the hippocampus and frontal lobes.

The inclusion criteria of the animal experiments studies were as follows: (1) Animals were modeled for heart failure or myocardial infarction (MI) in vivo experiment; (2) The researches pointed out the structure and function of the hippocampus and frontal lobes; (4) Languages were not restricted and the literatures should be published in the official journals.

The exclusion criteria were as follows: (1) Articles with incomplete information; (2) Reviews, meta-analysis, and corresponding/conference abstracts; (3) Comorbidities in clinical patients include neurological diseases such as stroke, Alzheimer's Disease (AD), protopathy of the brain and other neurological symptoms.

## 3. Study selection and data extraction

Titles and abstracts of all studies were assessed independently by two researchers (ZW.L. and Y.T.) according to the inclusion and exclusion criteria. Firstly, duplicate literature that came from different databases were removed from the initial results. Secondly, distinctly irrelevant literature were eliminated via titles and abstracts. Thirdly, we screened the full texts and finally confirmed the included studies.

Two authors (ZW.L. and Y.T.) individually extracted data from the included literature employing a standardized sheet prepared for this review. Data on clinical studies included clinical characteristics of the study population were collected, it consists of first author's name, year of publication, country, sample size, age, HF severity and comorbidities; cognitive function and psychological test; changes and damage in the hippocampus as well as relevant hippocampal region and changes of frontal lobes. As for animal researches, the basic information of included experimental studies was extracted, including animal species, sex, weight, age, sample size, etc. In addition, we also acquired the cognition and depression tests and characteristics of hippocampal and frontal lobes changes. Any disagreement was resolved by discussing and consulting with the corresponding authors (MJ.Z.).

## 4. Assessment of bias risk and study quality

Newcastle-Ottawa scales (NOS) was used to assess the quality of clinical studies based on three factors: the selection of research population, compatibility of the study groups, and measurement of exposure factors. Each study scored 0–9 points. Cochrane risk of bias tool was used to assesses clinical studies as low or high risk for the following forms of bias: selection, performance, detection, attrition, reporting, and other.

SYRCLE's risk of bias tool was applied to assess the quality of animal researches, a total of ten items of six projects were used as evaluation criteria including selection bias, performance bias, detection bias, attrition bias, reporting bias, and other sources of bias. Each item was evaluated as "high risk", "low risk", or "unclear" [12].

## 5. Summary analysis

Due to the high heterogeneity of the included literature, we only compared different trends and mechanisms in the hippocampus change both in clinical and animal researches. Therefore, a qualitative synthesis was adopted for this systematic review.

## Results

### 1. Overview of the studies

**1.1 Search results.**   The process of literature selection, including identification, screening, eligibility and included, are described in Fig 1. Of the 1036 articles in search of PUBMED, Embase, Web of Science, and Cochrane Library, 80 were excluded because of duplication, and 910 were excluded after reading titles and abstracts. Then 16 articles were excluded according to the full text and finally 30 studies (17clinical researches+13 animal studies) were included in this review (shown in Fig 1).

### 1.2 Characteristics of clinical researches

As shown in Table 1, among the included 17 clinical studies, seven studies were conducted in America and four studies were in Japan, while the remaining studies were based in Germany (2/17), Brazil (1/17), China (1/17), Australia (1/17) and Poland (1/17). There were ten case-control studies, three cross-sectional study, one prospective study, two cohort studies, and one

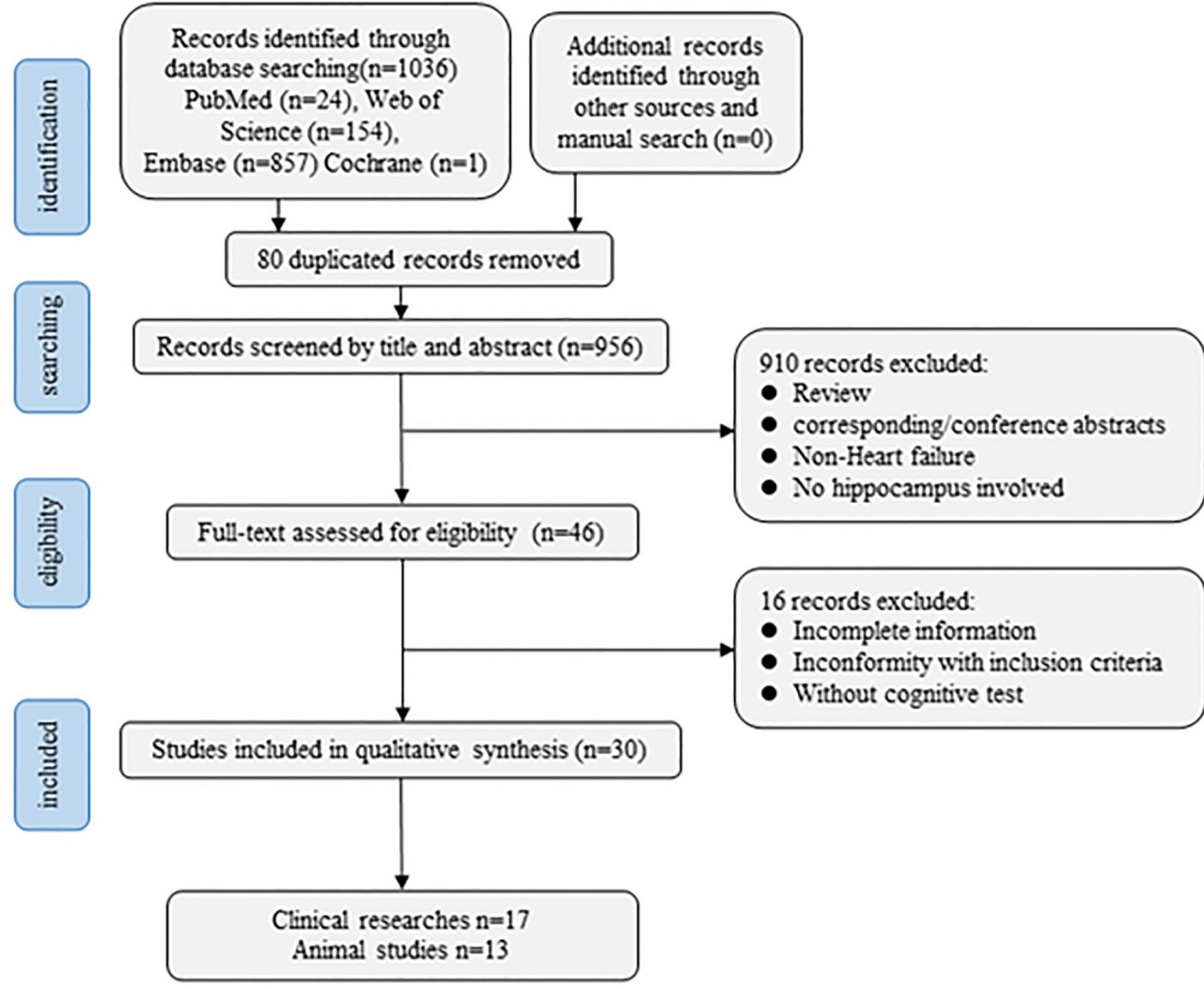

**Fig 1. Flowchart of the study selection process.**

**Table 1. Summary of clinical study characteristics.**

| Author, year | Country | Study design | Sample size | Mean age | LVEF (%) | NYHA class | NT-proBNP level(pg/ml) | Comorbidities | ACC/AHA Stages of HF |
|---|---|---|---|---|---|---|---|---|---|
| Roy, B 2017 [18] | America | case-control study | •Control group: 29 (healthy subjects) •HF group: 19 | 51.4 55.5 | Control:— HF:30.5±11.5 | II (80%) III (20%) | N/A | Hypertensive (12) atrial fibrillation (4) type 2 diabetes (5) | Stage C |
| Mueller, K 2020 [33] | Germany | case-control study | • Control group: 60 (healthy subjects) •NAD group: 22 •CAD- group: 20 •CAD+ group: 35 | 54.9 | Control:— NAD:62.5±5.4 CAD-:63.7±5.4 CAD+:47.2 ±11.7 | N/A | NAD:68.8 CAD-:66.7 CAD+:2758.8 | Arterial hypertension (50) Diabetes (18) Smoking (57) | Stage B |
| Pan, A 2013 [20] | America | two-group comparative study | •Control group: 50 (healthy subjects) •HF group: 17 | 50.6 54.4 | Control:— HF:28±7 | II ($n = 17$) | N/A | Type II diabetes (4) atrial fibrillation (2) hypertension (6) | Stage C |
| T AN I A C. T. F. A L V E S, 2006 [16] | Brazil | cross-sectional design | •Control group: 18 (healthy subjects) •MDD-HF group:17 •Non-depressed HF group: 17 | 72.8 76.0 73.7 | Controls: 73.4±4.2 MDD-HF: 35.5±7.6 Non-depressed HF:39.8±3.3 | II ($n = 18$) III ($n = 16$) | N/A | atrial fibrillation (5) cigarette smoking (12) diabetes (20) | Stage C |
| Suzuki, H 2016 [9] | Japan | case-control study | •Stage B group: 40 •Stage C group: 40 | 65.0 66.8 | Stage B: 59.6±14.7 Stage C: 43.1±17.5 | N/A | N/A | (StageB/StageC) Hypertension (52.5%/65%) Diabetes (35%/20%) Smoking (70%/57.5) | Stage B Stage C |
| Frey, A 2021 [15] | Germany | cohort study: follow up 1 and 3 years | 148 mild stable HF patients | 64.5 | Mean LVEF: 43.6±8.1 | I ($n = 41$) II ($n = 88$) III ($n = 19$) | N/A | Myocardial infarction (54.1%) Atrial fibrillation (19.6%) Hypertension (79.7%) Diabetes mellitus (29.1%) Renal dysfunction (35.8%) | Stage C |
| Suzuki, H 2020 [21] | Japan | cohort study: follow up 3.1 ± 0.5 years | 70 CHF patients | 65.0 | Mean LVEF: 51.3±16.9 | N/A | 107.5±127.5 | Ischemic heart failure (54.3%) Hypertension (64.3%) Diabetes (30%) | Stage B Stage C |
| Niizeki, T 2019 [29] | Japan | prospective study | 491 CHF patients | 84 | Mean LVEF: 52±15 | I ($n = 32$) II ($n = 214$) III ($n = 205$) IV($n = 40$) | 421 | Hypertension (151/ 286) Diabetes mellitus (43/105) Hyperlipidemia (74/ 149) Atrial fibrillation (94/160) | Stage C |
| Woo, M. A 2015 [30] | America | case-control study | •Control group: 34 (healthy subjects) •HF group: 17 | 52.3 54.4 | Control:— HF:28.3±6.8 | II (94%) III (6%) | N/A | N/A | Stage C |

(*Continued*)

**Table 1.** (Continued)

| Author, year | Country | Study design | Sample size | Mean age | LVEF (%) | NYHA class | NT-proBNP level(pg/ml) | Comorbidities | ACC/AHA Stages of HF |
|---|---|---|---|---|---|---|---|---|---|
| Yun, M 2020 [37] | China | case-control study | •Control group:55 (healthy subjects) •HF group: 102 | 56.3 | Control:— HF:15–37.3 | III--IV (n = 50) | >100 | Hypertension (36.3%) Diabetes (32.3%) Dyslipidemia (21.6%) Prior and current smokers (76.4%) | Stage C |
| Woo, M. A 2005 [32] | America | case-control study | •Control group:16 (healthy subjects) •HF group: 6 | 48 49 | N/A | III-IV (n = 6) | N/A | Sinus rhythm(n = 5) Chronic atrial fibrillation(n = 1) | Stage C |
| Wykrętowicz, A 2019 [31] | Poland | case-control study | •Control group:38 (healthy subjects) •HF group: 25 | 62 64 | Control: 64 HF:32 | N/A | N/A | Hypertension (n = 10) Diabetes (n = 6) | Stage C |
| Menteer, J 2010 [34] | America | case-control study | •Control group:7 (healthy subjects) •HF group: 7 | 12.9 | Control:-- HF:14±3 | II (n = 2) III (n = 3) IV(n = 2) | N/A | N/A | Stage C |
| Woo, M. A 2009 [35] | America | case-control study | •Control group:13 (healthy subjects) •HF group: 49 | 50.6 54.6 | Control: -- HF:28±7 | II (n = 4) | N/A | Sinus rhythm (85%) Atrial fibrillation (15%) | Stage C |
| Almeida, O. P 2012 [17] | Australia | cross-sectional study | •Control group:64 (healthy subjects) •HF group: 35 | 68.7 69.2 | Control: 68.1±5.2 HF:30.4±7.8 | I (31.4%) II (54.3%) III (14.3%) | Control: 59.4 ±70.5 HF:233.8 ±194.5 | Smokers (68.6%) | Stage C |
| Park, B 2016 [22] | America | case-control study | •Control group:53 (healthy subjects) •HF group: 27 | 53 | Control: -- HF:28.0±9.2 | II (100%) | N/A | N/A | Stage C |
| Ichijo, Y. 2020 [19] | Japan | cross-sectional study | •Control group:35 (healthy subjects) •HF group: 28 | 70.5 70.6 | Control: -- HF:24 | I (60%) II (40%) | Control: 50.8 HF:346.6 | Hypertension (60%) Diabetes mellitus (54%) Dyslipidemia (77%) Atrial fibrillation (31.4%) | Stage C |

Abbreviations: **HF** = heart failure; **NYHA** = New York Heart Association; **LVEF** = left ventricular ejection fraction; **Stage C patients** = who had past or current CHF symptoms; **Stage B patients** = who had structural heart disease but had never had CHF symptoms; **CAD+** = coronary artery disease with heart failure; **CAD-** = coronary artery disease with sufficient heart function; **NAD** = no abnormality detected; **z-scores** = refers to the measured value of a growth index of the tested population and reference to the average value of this index and the ratio of the overall standard deviation of this index. The z-score in patients with cardiac events was significantly higher than that in patients without cardiac events. **MDD-HF** = major depressive disorder-heart failure; **NT-proBNP** = N-terminal prohormone of brain natriuretic peptide.

two-group comparative study. The sample sizes of the 17 studies ranged from 14 to 491 and most of them included males and females. The mean average age ranged from 12.9 to 84. The severity of HF for patients was assessed by the New York Heart Association (NYHA) classification for HF, left ventricular ejection fraction (LVEF), and the N-terminal prohormone of brain natriuretic peptide (NT-proBNP). Thirteen studies included HF patients with NYHA II to IV, while four of them also contained patients with NYHA I. The remaining studies only reported the LVEF value of patients with a range of 14% to 51.3%. According to ACC/AHA (American College of Cardiology/American Heart Association) stages of HF [13], fourteen studies included Stage C HF patients, and two studies included both Stage B and C HF patients. One study included Stage B HF patients. Most of the studies showed the comorbidities of the subjects.

**Table 2. General characteristics information and cardiac function of experimental studies.**

| Author, year | Sex/Animal strain | Weight(g) or Age (month) | Sample size | Model (Methods) | Time | Cardiac function assessments |
|---|---|---|---|---|---|---|
| Shinoda, 2016 [38] | Male mice | 10 weeks | 15 | HF(TAC) | 6 weeks | FS, LVESD, LVEDD↓; |
| Lijun zhang, 2019 [39] | Male mice | 8–10 weeks | 30 | HF(LAD) | 6 weeks | LVEF<40%; BNP↑ |
| C. Liu, 2014 [14] | Male rats | 250–300 g | 18 | MI(LAD) | 6 hours | infarct size was 50.25 ± 1.85% |
| Frey, 2014 [24] | Male mice | 6–9 weeks | 29 | HF(LAD) | 8 weeks | FS↓; ESD, EDD↑; ESA,EDA↑ |
| Koji Ito, 2013 [66] | Male mice | 10 weeks | 30 | MI(LAD) | 1 or 4 weeks | LV dimensions↑; FS↓ |
| Y. Zhou, 2020 [40] | Female rat | 12 weeks | 14 | HF(LAD) | 8 weeks | LVIDd, LVIDs, EDV and ESV↑; EF, FS, IVSd, IVSs↓ |
| Yingbin Ge, 2020 [41] | Male mice | 20–25 g | 30 | HF(LAD) | 2 weeks | LVEF↓; LVFS↓ |
| Austin T. H. Duong, 2019 [25] | Male mice | 8 weeks | / | HF(LAD) | 8 weeks | LVIDd, (LVIDs) dimension↑; LVEF↓; LVFS↓ |
| Camilo Toledo, 2019 [10] | Male rat | 250 g | 24 | Volume overload HF (Surgical creation of an arteriovenous fistula) | 8 weeks | EDD, EDV, ESD,ESV↑; MABP ↓; EF no change |
| H. Suzuki, 2015 [36] | Male rat | 9 weeks | 53 | HF(LAD) | 16 weeks | FS ↓; LVDd, PWTd↑ |
| Kim Lee, 2017 [26] | Male rat | 250–300 g | 32 | HF(LAD) | 10 weeks | LVESD,LVEDD↑; LVEF↓ |
| Md Rezaul Islam, 2021 [28] | Male mice | 3 months | / | HF(LAD) | | LVESD,LVEDD↑ |
| Yang, T 2020 [27] | Male rat | 240g | / | HF(LAD) | 10 days; 60days | LVEF↓, LVFS↓ |

Abbreviations: **HF** = heart failure; **LAD** = ligating the LAD coronary artery; **TAC** = Transverse aortic constriction; **LV** = left ventricle; **FS** = fractional shortening; **LVESD** = Left Ventricular End Systolic Diameter; **LVEDD** = left ventricular end-diastolic diameter; **BNP** = Brain Natriuretic Peptide; **ESD** = end-systolic diameter; **EDD** = end-diastolic diameter; **ESA** = end-systolic area; **EDA** = end-diastolic area; **LVIDd** = left ventricular internal dimension-diastole; **LVIDs** = left ventricular internal dimension in systole; **EDV** = end-diastolic volume; **ESV** = end-systolic volume; **IVSd** = interventricular septal defect; **IVSs** = interventricular septal systole; **LVFS** = left ventricular fractional shortening; **MABP** = mean arterial blood pressure; **LVDd** = left ventricular diastolic dimension; **PWTd** = posterior wall thickness in diastole.

**1.3 Characteristics of animal studies and assessment of HF model.** Table 2 presents the data of experimental studies on authors, animal race, sex, age or weight, and sample size. Among the 13 experimental studies, 11 studies reported that the HF model was made by ligating the LAD coronary, and the surgery of transverse aortic constriction and arteriovenous fistula were made to build the HF model in the remaining two studies. The feeding time after surgery of all experimental studies ranged from 6 hours to 16 weeks. To evaluate the cardiac function of animals, thirteen studies presented a significant decrease in EF and FS, while there was no change in EF of one study [10]. In addition, there was only one study recording the infarct size (50.25 ± 1.85%) of the heart [14].

## 2. Qualitative analysis

The results of using NOS to evaluate the risk of bias and the method quality of 16 clinical studies were shown in Table 3. The mean score of NOS in the included clinical studies was 7.3 with the total scores ranging from 75% - 100%. One included clinical study was assessed by Cochrane Risk of Bias Tool and was found to be at low risk of bias (Table 4). According to the assessment of SYRCLE's ROB tool, the included experimental studies displayed unclear of their sequence generation, random outcome assessment, blinding methods and other sources of bias. Most studies were low risk in baseline characteristics, random housing, incomplete

**Table 3. Newcastle–Ottawa Scale (NOS) of included clinical studies.**

| Author | Year | Method | Sample Size | Score |
|---|---|---|---|---|
| Roy, B [18] | 2017 | case-control study | 48 | 8 |
| Mueller, K [33] | 2020 | case-control study | 137 | 7 |
| T AN I A C. E S [16] | 2006 | cross-sectional study | 52 | 7 |
| Suzuki, H [9] | 2016 | case-control study | 80 | 6 |
| Frey, A [15] | 2021 | cohort study | 148 | 8 |
| Suzuki, H [21] | 2020 | cohort study | 70 | 7 |
| Niizeki, T [29] | 2019 | prospective study | 491 | 8 |
| Woo, M. A [30] | 2015 | case-control study | 51 | 7 |
| Yun, M [37] | 2020 | case-control study | 117 | 6 |
| Woo, M. A [32] | 2005 | case-control study | 22 | 7 |
| Wykrętowicz, A [31] | 2019 | case-control study | 63 | 6 |
| Menteer, J [34] | 2010 | case–control study | 14 | 7 |
| Woo, M. A [35] | 2009 | case-control study | 62 | 8 |
| Almeida, O. P [17] | 2012 | case-control study | 155 | 7 |
| Park, B [22] | 2016 | case-control study | 80 | 6 |
| Ichijo, Y. [19] | 2020 | cross-sectional study | 63 | 7 |

outcome data, and selective outcome reporting. More details of the risk of bias of experimental studies were in Table 5.

## 3. Obvious cognitive dysfunction and depression in HF

In most of the clinical researches, the measurements used to evaluate cognitive function in HF patients varied, including MMSE, MoCA, TMT-A, TMT-B, GDS, BDI-II, WMS-R, HDS-R, CAMCOG, CVLT tests (more details shown in Table 6). Three studies reported the attention or executive function deficit [15–17] in HF patients and four studies presented memory impairment [9, 16, 18, 19]. Moreover, one study particularly described the poor abilities of language, remote memory, praxis, calculation, abstract reasoning, and perception subscales of HF patients [16]. While two studies believed there were no changes in cognition. In addition, several studies (*n* = 4) also proved the depressive symptoms in heart failure [9, 20–22]. However, there were six studies did not detect the cognitive function in HF patients.

In animal studies, global cognition of animals was evaluated by passive avoidance tasks, active avoidance tasks, tail suspension test, forced swim test, sucrose preference test, open field tests, long-term potentiation recording, elevated plus maze, Y-maze test, object-in-place memory task, object oddity perceptual task and Morris water maze task(*n = 13*) (more details shown in Table 7). The reduction of learning [23] and memory in HF animals existed in six studies, and memory impairment manifested in short-term recognition memory [24], OiP memory [25], spatial memory [10, 26, 27], hippocampus-dependent memory [28].

**Table 4. Cochrane Risk of Bias Tool of included clinical studies.**

| Author, year | Random Sequence Generation | Allocation Concealment | Blinding of Participants & Personnel | Blinding of Outcome Assessment | Incomplete Outcome Data | Selecting Reporting | Other Sources of Bias |
|---|---|---|---|---|---|---|---|
| Pan, A, 2013 [20] | low risk | low risk | low risk | low risk | low Risk | high Risk | high Risk |

**Table 5. SYRCLE's risk of bias tool of animal experiments.**

| Author, year | Sequence generation (Selection bias) | Baseline characteristics (Selection bias) | Allocation concealment (Selection bias) | Random housing (Performance bias) | Blinding (Performance bias) | Random outcome assessment (Detection bias) | Blinding (Detection bias) | Incomplete outcome data (Attrition bias) | Selective outcome reporting (Reporting bias) | Other sources of bias |
|---|---|---|---|---|---|---|---|---|---|---|
| Shinoda Y, 2016 [38] | unclear | unclear | unclear | low risk | unclear | unclear | unclear | low risk | high risk | unclear |
| Lijun zhang, 2019 [39] | unclear | low risk | unclear | low risk | unclear | unclear | unclear | low risk | low risk | unclear |
| C.Liu, 2014 [14] | unclear | low risk | unclear | low risk | unclear | unclear | unclear | high risk | low risk | unclear |
| Frey, A 2014 [24] | unclear | low risk | high risk | low risk | unclear | unclear | unclear | low risk | low risk | unclear |
| Koji Ito, 2013 [66] | unclear | unclear | unclear | unclear | unclear | unclear | unclear | unclear | high risk | unclear |
| Y. Zhou, 2020 [40] | unclear | unclear | high risk | low risk | unclear | unclear | unclear | unclear | low risk | unclear |
| Yingbin Ge, 2020 [41] | unclear | low risk | unclear | unclear | unclear | unclear | unclear | low risk | low risk | unclear |
| Austin T. H. Duong, 2019 [25] | unclear | low risk | unclear | low risk | unclear | unclear | unclear | unclear | low risk | unclear |
| Camilo Toledo, 2019 [10] | unclear | unclear | unclear | low risk | unclear | unclear | unclear | unclear | low risk | unclear |
| H. Suzuki, 2015 [36] | unclear | unclear | high risk | low risk | unclear | unclear | unclear | low risk | low risk | unclear |
| Kim, Lee, Kim, 2017 [26] | unclear | low risk | unclear | low risk | unclear | unclear | unclear | unclear | low risk | unclear |
| Md Rezaul Islam. 2021 [28] | unclear | low risk | unclear | low risk | unclear | unclear | unclear | low risk | high risk | unclear |
| Yang, T 2020 [27] | unclear | low risk | low risk | low risk | unclear | unclear | unclear | low risk | high risk | unclear |

## 4. Abnormalities of hippocampus in HF patients and animal models

**4.1 Differences in hippocampal volume of HF patients.** Five clinical studies (5/17) confirmed that the mean hippocampal volume decreased in HF patients compared to control groups [15, 20, 29–31]. Among these studies, three of them found significant hippocampal atrophy mainly reflected in the right hippocampus [15, 20, 30]. There were no studies about hippocampal volume in animal experiments. The detailed results are shown in Table 6.

**4.2 CBF alternations of the hippocampus in HF patients.** As illustrated in Table 6, several studies($n = 4$) indicated hippocampus showed lower CBF values in HF patients [9, 16, 18, 21]. However, in one study, CBF in the whole hippocampus of HF patients tended to be lower without statistical significance ($P = 0.279$), while CBF had a regional significant reduction, mainly reflected in the most posterior portion of the hippocampus [9]. Obviously, CBF

**Table 6. Characteristics of cognition and hippocampal changes in clinical studies.**

| Author, year | Cognitive test | Cognitive function | Method(s) for hippocampal change | Hippocampal damage | Hippocampal damage regions |
|---|---|---|---|---|---|
| Roy, B 2017 [18] | MoCA, Beck depression/anxiety inventory (BDI/BAI) | HF group: significant **decrease** in MoCA, BDI scores and delayed recall compared with control group. | MRI | HF group: significant **decrease** in cerebral blood flow (CBF) compared with control group. | Right and left side |
| Mueller, K 2020 [33] | Neuropsychological tests battery | No differences in attention, executive function and memory between groups. | MRI | CAD with HF group: gray matter density **reduced** significantly compared with LIEF group. | N/A |
| Pan, A 2013 [20] | BDI-II TMT-B | HF group: significant **increase** in BDI-II and TMT-B scores compared with control group. | MRI/Visual Assessment | HF group: significant difference showed in hippocampal atrophy compared with control group. | Right hippocampus |
| TAN I A C. T. F. A L V E S 2006 [16] | MMSE, CAMCOG | HF groups: significant **decrease** in MoCA and CAMCOG scores compared with healthy group. | 99mTc-SPECT | MDD-HF group: significant **decrease** in CBF compared with non-depressed HF group and healthy group. | Right posterior hippocampus/posterior para-hippocampal gyrus/Left anterior para-hippocampal gyrus/anterior hippocampus |
| Suzuki, H 2016 [9] | Psychological tests (MMSE,GDS,WMS-R, IM,DM) | Stage C HF patients: significant **increase** in GDS scores and **decrease** in IM and DM scores compared with control group. | MRI | Stage C HF patients: significant **decrease** in CBF compared with control group. | Posterior hippocampus postero-posterior hippocampus |
| Frey, A 2021 [15] | Psychological test battery | The intensity of attention **declined** in HF patients over 3years. | MRI | The mean hippocampal volume **declined** in HF patients over time. | Right side |
| | | No differences in selectivity of attention and working memory. | | | |
| Suzuki, H 2020 [21] | Psychological tests (GDS, WMS-R) | The GDS scores **increased** in HF patients. | MRI | The anterior hippocampal blood flow was negatively correlated with changes in PWT in HF patients. | anterior and posterior hippocampal |
| Niizeki, T 2019 [29] | HDS-R | The patients with cardiac events (z-scores high group): significant **decrease** in HDS-R scores | MRI | The patients with cardiac events: significant **increased** prevalence of hippocampal atrophy compared with patients without cardiac events | N/A |
| Woo, M. A 2015 [30] | N/A | N/A | MRI | HF group: significant decline in right hippocampal volume. | Right hippocampus; CA1 and CA3 region |
| Yun, M 2020 [37] | N/A | N/A | 18F-FDG PET/CT imaging | HF group: significant decrease in glucose metabolism of hippocampus | Right hippocampus |
| Woo, M. A 2005 [32] | N/A | N/A | MRI | HF group: significant decrease in gray matter of hippocampus | N/A |
| Wykrętowicz, A 2019 [31] | N/A | N/A | MRI | HF group: significant reduction of hippocampus volumes | N/A |
| Menteer, J 2010 [34] | N/A | N/A | MRI | HF group: significant gray matter loss in hippocampus | Right mid-hippocampus |

(*Continued*)

**Table 6.** (Continued)

| Author, year | Cognitive test | Cognitive function | Method(s) for hippocampal change | Hippocampal damage | Hippocampal damage regions |
|---|---|---|---|---|---|
| Woo, M. A 2009 [35] | N/A | N/A | MRI | HF group: significant higher T2 relaxation values(loss of gray and white matter) in hippocampus. | Right hippocampus |

Abbreviations: **Psychological test battery** include TAP, VVM2, WMSR, RET, H5PT tests(**TAP** = test battery of attentional processes; **VVM2** = the Visual and Verbal Memory Test; **WMS-R** = the Wechsler Memory Scale revised; **RET** = the Regensburger Word Fluency Test; **H5PT** = hamasch 5-point test revised). **Neuropsychological tests battery** include TMT-A, TAP,TMT-B, CVLT, ROCF (**TMT-A** = trail making test A; **TAP** = test battery of attentional processes; **TMT-B** = trail making test B; **RWT** = Regensburg word fluency test; **CVLT** = California verbal learning test; **ROCF** = Rey-Osterrieth complex figure test). **MoCA** = Montreal Cognitive Assessment test. **BDI-II** = the Beck depression inventory. **BAI** = the Beck anxiety inventory. **MMSE** = Mini-Mental State Examination. **CAMCOG** = Cambridge Mental Disorders of the Elderly Examination. **GDS** = Geriatric Depression Scale. **WMS-R** = the Wechsler Memory Scale-revised. **IM** = immediate memory. **DM** = a test of delayed memory. **PWT** = Posterior wall thickness; **HDS-R** = the Revised Hasegawa's Dementia Scale; BA27/30, the right posterior hippocampus and posterior para-hippocampal gyrus; BA28/34/35/36, left anterior para-hippocampal gyrus. **GMD** = gray matter density. **z-scores** = refers to the measured value of a growth index of the tested population and reference to the average value of this index and the ratio of the overall standard deviation of this index. The z-score in patients with cardiac events was significantly higher than that in patients without cardiac events.

decrease was also observed in anterior hippocampal gyrus of some studies(*n = 2*) [16, 21]. There were no studies about hippocampal CBF in animal experiments.

**4.3 Gray matter decrease in hippocampus in HF patients.** Other hippocampal injuries in HF were reduction of gray matter. Four studies [32–35] showed a significant GMD decrease in hippocampus and surprisingly, almost all of them reported this change in the right side of hippocampus except one study did not mention regional change. Similarly, there was one experimental study also observed a decrease in gray matter concentration in HF rats hippocampus [36]. The detailed results are shown in Table 6.

**4.4 Decrease in glucose metabolism of hippocampus in HF patients.** As illustrated in Table 6, one study [37] reported the significant decrease in glucose metabolism of hippocampus. Interestingly, this study also represented that this kind of changes exhibited in the right side of hippocampus.

**4.5 Hippocampal damages in HF animal model.** Clinical research about hippocampal damages in heart failure is just limited to phenomenological changes, and deeper potential molecular mechanisms should be explored through basic experimental researches. Among experimental studies, each study analyzed different genes or proteins with statistical significance. The detailed results are shown in Table 7.

## 5. Abnormalities of frontal lobe in HF patients and animal models

There were four clinical studies and one experimental study involved the changes of frontal lobe in heart failure. The detailed results are shown in Tables 7 and 8. Two studies also reported the loss of gray matter of frontal lobe in HF [17, 33]. The other two studies showed significant decreased functional connectivity and mean oxyhemoglobin concentrations in frontal gyrus, respectively [19, 22].

## Discussion

Heart failure patients shows obvious cognitive impairment or depression, which increases the mortality and rehospitalization rate of HF patients. The hippocampus and frontal lobe were considered to be the most important brain regions for cognitive flexibility (such as cognition

**Table 7. Characteristics of cognitive function and hippocampal and frontal lobe changes of HF rats in included experimental studies.**

| Author, year | Behavior tests | Cognitive function | Changes of hippocampus | Mechanisms or intervention |
|---|---|---|---|---|
| Shinoda Y, 2016 [38] | TST; SPT forced swim test; | Depression-like behaviors | Expression of σ1-receptor in the CA1 region and dentate gyrus↓ | Increases in plasma corticosterone (CORT) levels; Corticosteroids |
| Lijun zhang, 2019 [39] | SPT, OFT | Depressive behaviors | Expression of 5-HT, 5-HT receptor↑ | Ginkgo biloba Extract |
| C. Liu, 2014 [14] | LTP | Suppression of long-term potentiation | Levels of MDA and H2O2↑; Cu/Zn-SOD activity↓; NR2B expression↓ | Activation of p-Akt/Akt |
| Frey, 2014 [24] | SPT,EPM, LDB, OFT; OR | Depression and anhedonia; reduced exploratory behavior; deficits in stress-coping; less habituation to new environment; deficits in short-term recognition memory | Transcriptional up-regulation of HIF-3α; retinoid-related orphan receptor-alpha↑; gene expression of Kif5b and Gabrb2↓ | Serotonin system; differences of RNA expression of several hippocampal regions |
| Koji Ito, 2013 [66] | TST; Y-maze test | Decreased spontaneous alternation | Expression of sigma-1 receptor↓ | Sigma-1 receptor |
| Y. Zhou, 2020 [40] | EPM, LDB, OFT | Exhibiting anxiety-like behavior | KDM6B↓; SIRT1↑; IL-1β, Bax, cleaved-caspase 3 proteins↑ | MI-induced neuro-inflammation and neuronal apoptosis; downregulation of KDM6B but upregulation of SIRT1 signaling |
| Yingbin Ge 2020 [41] | OFT, SPT | Depression-like behaviors | Expression of 5-HT↑; IL-1β↑ | Ginkgolide B attenuates myocardial infarction-induced depression-like behaviors |
| Austin T. H. Duong, 2019 [25] | OiP, Object oddity perceptual task | Impairment in OiP memory; object oddity discrimination in HF remains intact | Basal dendrite length increases↑; differences predominantly mapped to metabolic pathways | Core circadian mechanism |
| Camilo Toledo, 2019 [10] | MWM | Learning and memory impairment; spatial memory deficits | Expression levels of active β-catenin, pGSK-3β↓ | Wnt/β-catenin signaling↓ |
| H. Suzuki, 2015 [36] | 24-hour LAM | Depressive symptoms | Decrease in gray matter concentration, neurogenesis and neurite outgrowth; increase in the number of astrocytes | -- |
| Kim, Lee, 2017 [26] | Morris water maze task | Impairment of spatial memory | Cell death in the area of the hippocampus | Ang II receptor mediated cell death. |
| Md Rezaul Islam, 2021 [28] | OFT | Impaired hippocampus-dependent memory consolidation | Down-regulation of hippocampal genes | Reduced neuronal H3K4 methylation |
| Yang, T 2020 [27] | Morris water maze task | Impairment of spatial memory | Brain glucose metabolism of frontal cortex significantly lower in 60 days HF rats and higher in 10 days HF rats. | / |

Abbreviations: **OFT** = open field tests; **SPT** = Sucrose preference test; **TST** = Tail suspension test; **LDB** = light dark box; **EPM** = elevated plus maze; **OiP** = Object-in-place memory task; **LTP** = Long-term potentiation recording; **EPM** = elevated plus maze; **OR** = object recognition; **MWM** = Morris Water Maze; **LAM** = locomotor activity measurement.

and mood) during the HF stage. However, there are scarcely any systematic reviews about changes of the hippocampus and frontal lobe in HF patients and animal models.

## 1. Statement of key findings

This systematic review synthesized available studies about changes of hippocampus and frontal lobe in HF and finally included 30 studies (17 clinical researches and 13 animal studies). This review scientifically summarized the cognitive dysfunction and depressive symptoms in HF which reflected in poor abilities of attention, executive function, learning and memory, and anhedonia, reduced exploratory or anxiety-like behavior in both HF patients and animals. Importantly, this review integrally concluded changes of the hippocampus and frontal lobe in

**Table 8. Characteristics of cognition and changes of frontal lobe in clinical studies.**

| Author, year | Cognitive test | Cognitive function | Method(s) for frontal lobe change | frontal lobe damage | frontal lobe damage regions |
|---|---|---|---|---|---|
| Almeida, O. P 2012 [17] | CAMCOG, CVLT, digit coding/copying | HF group: significant **decrease** in CVLT and digit coding scores compared with control group. | MRI | HF group: significant loss of gray matter in frontal gyrus compared with control group. | Right inferior/middle/ precentral frontal gyrus; left middle frontal gyri |
| Park, B 2016 [22] | MoCA, BDI, BAI | HF group: significant **decrease** in MoCA, BDI, BAI scores compared with control group. | MRI, brain network analysis | HF group: significant decreased functional connectivity in frontal gyrus. | N/A |
| Ichijo, Y. 2020 [19] | VFT, CES-D, STAI, MMSE | HF group: significant **increase** in VFT, STAI and MMSE scores compared with control group. | Near-Infrared Spectroscopy | HF group: significant lower mean oxyhemoglobin concentrations of frontal region compared with control group. | N/A |
| Mueller, K 2020 [33] | Neuropsychological tests battery | No differences in attention, executive function and memory between groups. | MRI | CAD with HF group: gray matter density **reduced** significantly in whole frontal cortex compared with LIEF group. | N/A |

Abbreviations: **CAMCOG** = Cambridge Mental Disorders of the Elderly Examination. **CVLT** = California verbal learning test; **MoCA** = Montreal Cognitive Assessment test. **BDI-II** = the Beck depression inventory. **BAI** = the Beck anxiety inventory. **MMSE** = Mini-Mental State Examination. **VFT** = verbal fluency task; **CES-D** = Center for Epidemiologic Studies Depression Scale. **STAI** = State-Trait Anxiety Inventory.

HF patients, and these changes mainly showed in volume atrophy, decreased CBF, reduced gray matter and glucose metabolism. Moreover, hippocampal damages of HF patients were mainly observed in the right side. According to the animal studies, the results showed the inflammation, synaptic abnormalities, and neurotransmitter disorders of hippocampus and frontal lobes in HF animal model, which complemented the clinical observations by providing potential mechanistic explanations of the changes in the hippocampus and frontal lobes. To our knowledge, this is the first systematic review to focus on the hippocampal and frontal damages in HF patients and animal model.

## 2. Obvious cognitive impairment and depression exhibited in HF

In this systematic review, we found out heart failure may be considered as an important reason for cognitive dysfunction and depression. Several studies included in our results represented that there was not an obvious relationship between cognitive impairment caused by HF and comorbidities (such as diabetes and CAD) [15, 29, 33]. Moreover, the experimental studies also demonstrated similar results. In the HF model without any confounding factors, animals all showed obvious cognitive impairment. These results suggested that heart failure may be one of the factors leading to cognitive impairment. However, there were two clinical studies that showed paradoxical results that there was no difference in attention, working memory, or executive function [15, 33]. The most principal reason was the differences in the range and specificity of the instruments used to assess cognition. The MoCA or MMSE was the instrument most studies were willing to preferring. For another, the heart failure severity of patients in these two studies was lighter, which may influence the cognitive results. Coincidentally, in our systematic review, we also found out the presence of cognitive impairment in an experimental model of HFpEF [10]. However, according to our searching results, we cannot judge the difference in cognitive impairment between the HFpEF and HFrEF models. Finally, most of our studies [10, 28] suggested that decreased cognitive performance observed in HF rats may dependent on hippocampally-dependent mechanisms. Depressive symptoms were also found to be prevalent in HF patients in this systematic review [9, 18, 20]. Similarly to clinical studies, depression-like behaviors were also found in HF animals in most of the experimental

studies($n = 6$) [24, 36, 38–41]. The previous study represented that frontal dysfunction has been observed in patients with depression, which may be further associated with cognitive impairment [42].

Moreover, cognitive impairment and depression in HF may be related to brain-derived neurotrophic factor (BDNF). BDNF plays a crucial role in memory formation and the hippocampus contains a high concentration of it [43]. Some studies reported that lower blood BDNF levels were associated with a higher incidence of dementia and depression in CHF patients [44, 45]. While since BDNF could be produced in the skeletal muscle, another research suggested that the decrease in serum BDNF levels may be due to the physical inactivity in HF patients [46]. Therefore, we believed that further studies are needed to investigate BDNF levels in exosomes derived from neurons, rather than plasma in HF patients, which could better-reflected abnormalities in the brain.

## 3. Hippocampal and frontal damage alterations--and possible mechanism

**3.1 Reduction in CBF.** In this review, the results represented that the CBF of the hippocampus decreased in HF for regional significant reduction not only in the posterior hippocampus but also in the anterior hippocampal gyrus. The hippocampus has different functions and neuroanatomy in the different subregions of the anterior-posterior axis [47]. The posterior hippocampus plays a major role in cognitive functions and memory, while the anterior hippocampus performs emotion and stress. The hippocampus is a very vascular structure and susceptive to changes in blood flow and hypoxemia [48]. It has been reported that reduced CBF in the hippocampus may possibly associate with poorer cognitive function and depressive symptoms in AD [49].

Some potential pathophysiology mechanisms may contribute to decreased regional CBF in HF. The possible mechanism for brain damage in HF patients may be due to cardiac dysfunction and decreasing cardiac index. The reduction of low cardiac output is may lead to ischemia which affects both vasculature and endothelial function. Altered endothelial function can result in abnormal cerebral autoregulation, which is mainly reflected in decreased CBF in HF patients. Cerebral autoregulation maintains a stable CBF over a wide range of mean arterial pressure and can be affected by the renin-angiotensin system [50], which is a central neurohormonal response to control cardiovascular and renal function. Therefore, B-type natriuretic peptide (BNP) neurohormones, which is released as a response to increased ventricular wall stretch and marker of the severity of HF [51], may cause a reduction of CBF. The high levels of BNP may result in a more active neurohormonal system in HF patients, and this may cause the distortion of brain autoregulation [51]. Thus, the hippocampus might be especially prone to damage caused initially by a disruption of blood flow. In addition, central venous pressure (CVP) represents the pressure of the thoracic vena cava near the right atrium, and patients with HF often accompany increased CVP [52]. While an increase in CVP can affect brain oxygen and capacity, causing a decrease in oxygen saturation and affecting cognitive function. Of note, one study represents that the influence of CVP on cerebral perfusion was most pronounced in low arterial flow states [53]. Moreover, variable degrees of carotid obstruction in HF might have contributed to the CBF reductions [54]. The abnormal hippocampal CBF in CHF patients may be reversible. Furthermore, one other possibility is that reduction of CBF in HF patients may be influenced by cerebrovascular reactivity. The previous study evaluated $CO_2$-reactivity and blood flow velocities of the middle cerebral arteries in CHF patients and demonstrated the impairment of cerebrovascular reactivity [55]. Lower $CO_2$ contributes to reduced cerebral arteriolar dilatation, which leads to reduced CBF [56]. And the decrease in cerebrovascular reactivity had a relationship with the decline in cardiac function.

**3.2 Reduction in volume.** The results in our systematic review demonstrated that the right hippocampus was significantly atrophied in HF patients. Previous researches have indicated that the right hippocampal functionality may be impaired in depression [57, 58]. The reason why brain atrophy occurred in the right hippocampus not the left side may be that the hippocampus exhibits lateralization [59]. This trend may also reflect preferential right-sided activation for sympathetic regulation, thus being more metabolically demanding of perfusion, which may be unmet by compromised cerebral autoregulation in HF. It was also consistent with our searching results of changes of CBF in hippocampus of HF patients.

**3.3 Decreased gray matter density and glucose metabolism.** In this systematic review, the gray matter density and glucose metabolism obviously decreased in the right hippocampus in HF, and one study demonstrated this regional GM was positively correlated with EF and negatively with NT-proBNP [33]. On the other hand, gray matter density was also reduced in the frontal lobes of HF patients, and glucose metabolism of chronic HF rats also declined in the frontal lobes. It has been shown that vascular dementia (VD) patients had greater deficits in frontal–executive functions, verbal fluency, attention, and motor function when compared to Alzheimer's disease (AD) of similar severity [60]. Therefore, we believe that cognitive impairment in patients with HF may be similar to vascular dementia to some extent.

Several previous studies suggested that the brain gray matter density was related to cardiovascular impairment [17, 61]. Gray matter, an important part of the central nervous system, was consisted of a large number of neuronal bodies and their dendrites. As in our previous study [27], we also found out that the number of neurons and synapses significantly decreased and Nissl bodies disappeared in HF rats. In other words, our previous studies indirectly proved the decreased gray matter density of HF patients in clinical.

These decreases in volume, gray matter density, and glucose metabolism of the hippocampus and frontal lobes may be related to low cardiac output and cerebral hypoxia. As discussed earlier in this review, low cardiac output in HF patients results in ischemia, which affects cerebral autoregulation and reduces cerebral blood flow. Subsequently, this decrease in CBF can contribute to a hypoxic environment and a shortage of glucose, leading to neuronal and glial necrosis in the hippocampus or frontal lobes [62]. It is likely that neuronal degeneration in a cerebral hypoxic environment is a slower process and may be due to a combination of excitotoxicity, producing necrosis, and apoptosis. In addition, the decrease in the mean oxyhemoglobin concentration in the frontal region may also contribute to brain structural abnormalities.

**3.4 Potential pathological mechanisms of hippocampal and frontal damages in HF animal model the exact mechanism.** This systematic review generalizes the molecular processes and gene expression of the hippocampus in the HF model. Firstly, that HF may induce neuroinflammation and cell apoptosis in the hippocampus. Several studies in our results demonstrated that the levels of IL-1β, Bax, cleaved-caspase 3 proteins increased in the hippocampus of HF rats [40, 41, 63], which may have a relationship with Ang II receptor and upregulation of SIRT1. Meanwhile, we also found two studies that suggested the expression of the sigma-1 receptor in the hippocampus was reduced in the heart failure model. Brain sigma-1 receptor (σ1R) plays a major role in inflammation, neurite outgrowth or Ca2+ signal regulation, and is reported as the key molecule of the pathogenesis in depression or cognitive impairment [64]. Secondly, HF animals were used to cause the alternations of neurotransmitters and their receptors in the hippocampus. As shown in our results, the levels of 5-HT increased in the hippocampus in HF rats in our results, which contributed to the development of depression. Moreover, the expression of NR2B protein, a major functional component of the hippocampal NMDA receptors, was obviously reduced in the MI model group. Thirdly, molecular mechanisms involving synaptic changes also were observed in HF model. One included experimental

study indicated the reduction of the Wnt signaling pathway in the hippocampus. Wnt signaling pathway has been largely implicated in the regulation of synaptic assembly, neurotransmission and synaptic plasticity in the adult nervous system [65]. Lastly, from the gene expression, downregulation in gene expression of Kif5b and Gabrb2 were observed in HF, which were responsible for axonal vesicular transport, and regulation of synaptic transmission, respectively. Our results suggested that hippocampal gene expression changes were observed in mice suffering from HF. One of the included experimental studies reported the decreased levels of the euchromatin mark H3K4me3 in the hippocampus. While the proper neuronal H3K4me3 is essential for memory consolidation in previous research.

## 4. Limitations

A major limitation of the included clinical studies reviewed is that the potential confounders for cognitive dysfunction could influence the results of these studies, including education level, without dividing into groups according to HFpEF or HFrEF, and diabetes. The other limitation is that the feeding time after MI surgery is included experimental studies is various and ranged from 6 hours to 16 weeks, which may have an influence on the results of cognition and hippocampal changes in the HF model. Finally, this review only focused on hippocampus while other putative brain structures that should be involved in future research, such as frontal lobes, amygdala, locus coeruleus, hypothalamus.

## 5. Implications for future researches

Cognitive impairment or depression in HF patients will induce the poor ability of self-care and poor quality of life in patients, while the hippocampal damages may provide a structural and functional explanation for the disabilities in cognition and depression. These indicated that the heart and brain are interconnected and interact with each other pathologically. And unfortunately, few studies are focusing on the effect of therapeutic interventions specifically for HF on cognitive function. This systematic review identifies underlying mechanisms of the hippocampus for cognitive disorders or depression in HF, which may provide treatment strategies and targets for intervention. In addition, it also could present data for future basic researches and further strengthen the notion that brain and heart function are tightly linked.

## Conclusions

A total of 30 literature studies were included to review the hippocampal and frontal changes in HF patients and animal models. Cognitive dysfunction in HF patients and animal models mainly showed poor abilities of attention, executive function, learning and memory, and anhedonia reduced exploratory or anxiety-like behavior. Hippocampal injuries of HF patients were shown in the reduction of volume, CBF, glucose metabolism, and gray matter, which were mainly observed in the right hippocampus. The frontal damages of HF patients were only in reduced gray matter and have no differences between the right and left sides. The included HF animal model studies demonstrated the inflammation and apoptosis, synaptic reduction, and neurotransmitter disorders of the hippocampus and frontal lobes, which contributed to the loss of gray matter and volume. The graphical abstract was shown in Fig 2. The results of HF animal model studies complemented the clinical observations by providing potential mechanistic explanations of the changes in the hippocampus and frontal lobes. Finally, this systematic review provided data and therapeutic targets for the prevention and treatment of cognitive impairment and depression after heart failure.

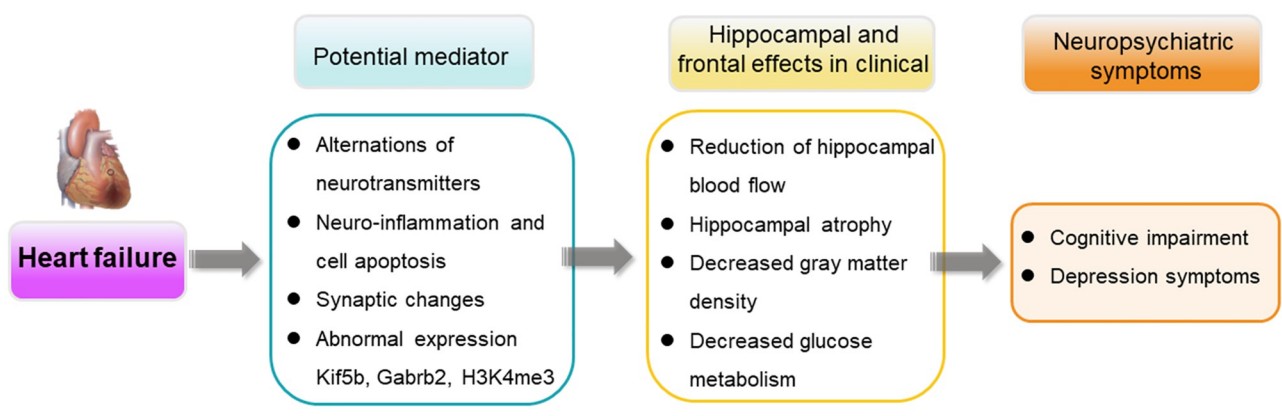

**Fig 2. Graphical abstract.**

## Supporting information

**S1 Checklist. PRISMA 2020 checklist.**
(DOCX)

**S1 File.**
(DOCX)

## Author Contributions

**Conceptualization:** Shiqi Chen, Xiangning Cui, Mingjing Zhao.

**Data curation:** Baofu Wang, Yang Li, Jingjing Yang, Xiaoxiao Wu.

**Investigation:** Mingjing Zhao.

**Methodology:** Lei Wang, Yangyang Jiang, Tong Li, Weiting Cheng.

**Project administration:** Mingjing Zhao.

**Visualization:** Mingjing Zhao.

**Writing – original draft:** Ziwen Lu, Yu Teng.

**Writing – review & editing:** Ziwen Lu, Yu Teng, Mingjing Zhao.

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
