## [Decision Letter · Decision Letter 0]

20 Sep 2022

PONE-D-22-23143Changes of hippocampus and frontal lobes in heart failure patients and animal models with cognitive impairment or depression: a systematic reviewPLOS ONE

Dear Dr. Zhao,

Thank you for submitting your manuscript to PLOS ONE. After careful consideration, we feel that it has merit but does not fully meet PLOS ONE’s publication criteria as it currently stands. Therefore, we invite you to submit a revised version of the manuscript that addresses the points raised during the review process.

We look forward to receiving your revised manuscript.

Kind regards,

Yoshihiro Fukumoto

Academic Editor

PLOS ONE

Journal Requirements:

5. Please ensure that you refer to Figure 2 in your text as, if accepted, production will need this reference to link the reader to the figure.

6. We note you have included a table to which you do not refer in the text of your manuscript. Please ensure that you refer to Table 4 in your text; if accepted, production will need this reference to link the reader to the Table.

Reviewers' comments:

Reviewer's Responses to Questions

**Comments to the Author**

1. Is the manuscript technically sound, and do the data support the conclusions?

Reviewer #1: Yes

Reviewer #2: Yes

2. Has the statistical analysis been performed appropriately and rigorously? 

Reviewer #1: N/A

Reviewer #2: I Don't Know

3. Have the authors made all data underlying the findings in their manuscript fully available?

Reviewer #1: Yes

Reviewer #2: Yes

4. Is the manuscript presented in an intelligible fashion and written in standard English?

Reviewer #1: Yes

Reviewer #2: Yes

5. Review Comments to the Author

Reviewer #1: The authors provide a review of previous reports of hippocampal damage in patients with chronic heart failure. This review was not aimed to broadly evaluate central nervous system changes in HF, but instead they focus this review on the hippocampus, which indeed is an important affected region of the brain in chronic heart failure.

The review goes on to VERY briefly discuss changes in the frontal lobes, but without also discussing changes in other areas of the brain such as the cingulate or the insular cortex. In my opinion, this review should focus on the hippocampus exclusively.

The visual abstract graphic seems to summarize how pathophysiology of heart failure may lead to the changes described in the manuscript, including vascular, gray matter, and metabolic changes, but the authors do not spend adequate time discussing the pathophysiology that may be causing the changes, except to simply explain that the hippocampus is metabolically active and therefore potentially more susceptible to damage than other areas. Issues such as impaired cardiac reserve with stress, elevated CVP and its relationship to cerebral perfusion pressure, and mechanisms of cerebral auto regulation are important considerations to brain function and are not discussed in adequate detail in the discussion.

Further, within the long list of authors for this meta-analysis, there does not appear to be any neurologist or neurophysiologist.

I believe that this manuscript deserves to be published after it is further refined.

Reviewer #2: The authors conducted a systematic review for abnormalities of the hippocampus and frontal lobe in heart failure (HF). The unique feature of this study is including both clinical and animal studies, which seem to be interesting. My comments are attached below.

1. Title page: "Changes" cannot be applied to all papers included in this study. This is because many of the included studies are cross-sectional and were not able to assess "changes", which include a meaning of causality. Words such as " abnormalities" or "differences" are more suitable in this case. Please modify "change(s)" to other words in this context.

2. Page 1-2: Abbreviations may be shown in order of earlier appearance. However, the alphabetical order would be more comprehensive.

3. Page 7-8: In 1.2 Characterstics of clinical researches, please also mention Stage classification of heart failure (PMID: 35379504).

4. Page 14-15: In 2. Obvious cognitive impairment and depression exhibitied in HF, please also discuss an association of brain-derived neurotrophic factor (BDNF) with cognitive impairment and depression in HF patients (PMID: 24029660; 28508502).

5. Page 15-16: In 3.1 Reduction in CBF, please also discuss a impairment in cerebrovascular reactivity in HF patients (PMID: 10666355).

6. Page 16-17: In 3.2 Reduction in volume or 3.3 Decreased gray matter density and glucose metabolism, please discuss an association of cerebral hypoxia with hippocampal damage as a mechanism of differences in grey matter volume and/or density in HF patients (e.g. hippocampal damage in cardiac arrest [PMID: 20101036; 23558096]; association of lower cardiac output with lower brain volume/density, including the hippocampus [PMID: 20679552; 29314453]).

6. PLOS authors have the option to publish the peer review history of their article (what does this mean?). If published, this will include your full peer review and any attached files.

Reviewer #1: No

Reviewer #2: No

---

## [Author Response · Author response to Decision Letter 0]

20 Oct 2022

Response to editors and reviewers

“1. Please ensure that your manuscript meets PLOS ONE's style requirements, including those for file naming. The PLOS ONE style templates can be found at https://journals.plos.org/plosone/s/file”

Response: Thank you for your comments.

As for this question, we have ensured our manuscript met PLOS ONE's style requirements.

“2. We note that the grant information you provided in the ‘Funding Information’ and ‘Financial Disclosure’ sections do not match. When you resubmit, please ensure that you provide the correct grant numbers for the awards you received for your study in the ‘Funding Information’ section.”

Response: Thank you for your comments.

As for this question, we have recorrected it.

“3. PLOS requires an ORCID iD for the corresponding author in Editorial Manager on papers submitted after December 6th, 2016. Please ensure that you have an ORCID iD and that it is validated in Editorial Manager. To do this, go to ‘Update my Information’ (in the upper left-hand corner of the main menu), and click on the Fetch/Validate link next to the ORCID field. This will take you to the ORCID site and allow you to create a new iD or authenticate a pre-existing iD in Editorial Manager. Please see the following video for instructions on linking an ORCID iD to your Editorial Manager account: https://www.youtube.com/watch?v=_xcclfuvtxQ”

Response: Thank you for your comments.

As for this question, we have provided ORCID iD.

“4. Please include your full ethics statement in the ‘Methods’ section of your manuscript file. In your statement, please include the full name of the IRB or ethics committee who approved or waived your study, as well as whether or not you obtained informed written or verbal consent. If consent was waived for your study, please include this information in your statement as well.”

Response: Thank you for your comments.

As for this question, we did not require an ethics statement. We have recorrected it to “N/A”.

“5. Please ensure that you refer to Figure 2 in your text as, if accepted, production will need this reference to link the reader to the figure.”

Response: Thank you for your comments.

As for this question, we have referred to Figure 2 in our text (lines 20 of page 22).

“6. We note you have included a table to which you do not refer in the text of your manuscript. Please ensure that you refer to Table 4 in your text; if accepted, production will need this reference to link the reader to the Table.”

Response: Thank you for your comments.

As for this question, we have referred to Table 4 in our text (line 21 of page 12).

Reviewer #1

“The review goes on to VERY briefly discuss changes in the frontal lobes, but without also discussing changes in other areas of the brain such as the cingulate or the insular cortex. In my opinion, this review should focus on the hippocampus exclusively.”

Response: Thank you for your assessment and comments.

As for this question, we have added the discussion of frontal lobes (lines 3-5 of page16; lines 2-8,16-22 of page 19). Because we focused on brain regions that showed abnormalities after HF in both the clinical researches and animal experiment studies. However, there was few animal experiment studies shown difference in the cingulate or the insular cortex. Therefore, we only focused on changes in the frontal lobe and hippocampus.

“The visual abstract graphic seems to summarize how pathophysiology of heart failure may lead to the changes described in the manuscript, including vascular, gray matter, and metabolic changes, but the authors do not spend adequate time discussing the pathophysiology that may be causing the changes, except to simply explain that the hippocampus is metabolically active and therefore potentially more susceptible to damage than other areas. Issues such as impaired cardiac reserve with stress, elevated CVP and its relationship to cerebral perfusion pressure, and mechanisms of cerebral auto regulation are important considerations to brain function and are not discussed in ade quate detail in the discussion.”

Response: Thank you for your careful review and comments. 

1. As for the question “the authors do not spend adequate time discussing the pathophysiology that may be causing the changes, except to simply explain that the hippocampus is metabolically active and therefore potentially more susceptible to damage than other areas”, we have enriched and expanded the discussion of pathophysiology caused damages in brain (lines 6-22 of page 17; lines 1-10 of page 18; lines 16-22 of page 19; lines 1-3 of page 20).

2. As for the question “the Issues such as impaired cardiac reserve with stress, elevated CVP and its relationship to cerebral perfusion pressure, and mechanisms of cerebral auto regulation are important considerations to brain function and are not discussed in ade quate detail in the discussion”, we have added them in our manuscript (lines 6-22 of page 17; lines 1-2 of page 18; lines 16-22 of page 19).

“Further, within the long list of authors for this meta-analysis, there does not appear to be any neurologist or neurophysiologist.”

Response: Thank you very much for your comments.

As for the question, the researchers in our team have been engaged in cardiovascular and cerebrovascular related research for a long time.

Reviewer #2

“1. Title page: "Changes" cannot be applied to all papers included in this study. This is because many of the included studies are cross-sectional and were not able to assess "changes", which include a meaning of causality. Words such as " abnormalities" or "differences" are more suitable in this case. Please modify "change(s)" to other words in this context.”

Response: Thank you for your careful review and comment! 

As for this question, we have modified it in our manuscript (title page; lines 9-10 of page 11; lines 13,19 of page 12).

“2. Page 1-2: Abbreviations may be shown in order of earlier appearance. However, the alphabetical order would be more comprehensive.”

Response: Thank you for your assessment of our article!

As for the question, we have modified it in our manuscript (lines 18-22 of page 1; lines 1-18 of page 2).

“3. Page 7-8: In 1.2 Characterstics of clinical researches, please also mention Stage classification of heart failure (PMID: 35379504).”

Response: Your advice is appreciated and has been valuable in improving the quality of our manuscript!

As for the question, we have added it in our manuscript (lines 11-14 of page 8; table 1)

“4. Page 14-15: In 2. Obvious cognitive impairment and depression exhibitied in HF, please also discuss an association of brain-derived neurotrophic factor (BDNF) with cognitive impairment and depression in HF patients (PMID: 24029660; 28508502).”

Response: Thanks for your comments and assessment of our article!

As for the question, we have added it in our discussion (lines 6-15 of page 16).

“5. Page 15-16: In 3.1 Reduction in CBF, please also discuss a impairment in cerebrovascular reactivity in HF patients (PMID: 10666355).”

Response: Thank you for your advises about our article. It is valuable in improving the quality of our manuscript. As for the question, we have added it in our discussion (lines 4-10 of page 18).

“6. Page 16-17: In 3.2 Reduction in volume or 3.3 Decreased gray matter density and glucose metabolism, please discuss an association of cerebral hypoxia with hippocampal damage as a mechanism of differences in grey matter volume and/or density in HF patients (e.g. hippocampal damage in cardiac arrest [PMID: 20101036; 23558096]; association of lower cardiac output with lower brain volume/density, including the hippocampus [PMID: 20679552; 29314453]).”

Response: Thank you for your advises about our article. 

As for the question, we have added it in our discussion (lines 16-22 of page 19, lines 1-3 of page 20).

---

## [Decision Letter · Decision Letter 1]

16 Nov 2022

Abnormalities of hippocampus and frontal lobes in heart failure patients and animal models with cognitive impairment or depression: a systematic review

PONE-D-22-23143R1

Dear Dr. Zhao,

We’re pleased to inform you that your manuscript has been judged scientifically suitable for publication and will be formally accepted for publication once it meets all outstanding technical requirements.

Kind regards,

Yoshihiro Fukumoto

Academic Editor

PLOS ONE

Additional Editor Comments (optional):

Reviewers' comments:

Reviewer's Responses to Questions

**Comments to the Author**

1. If the authors have adequately addressed your comments raised in a previous round of review and you feel that this manuscript is now acceptable for publication, you may indicate that here to bypass the “Comments to the Author” section, enter your conflict of interest statement in the “Confidential to Editor” section, and submit your "Accept" recommendation.

Reviewer #1: All comments have been addressed

Reviewer #2: All comments have been addressed

2. Is the manuscript technically sound, and do the data support the conclusions?

Reviewer #1: Yes

Reviewer #2: Yes

3. Has the statistical analysis been performed appropriately and rigorously? 

Reviewer #1: Yes

Reviewer #2: N/A

4. Have the authors made all data underlying the findings in their manuscript fully available?

Reviewer #1: Yes

Reviewer #2: (No Response)

5. Is the manuscript presented in an intelligible fashion and written in standard English?

Reviewer #1: Yes

Reviewer #2: Yes

6. Review Comments to the Author

Reviewer #1: My comments and questions have been addressed. Thank you and nice work on this meta-analysis! ......

Reviewer #2: The authors have addressed all of my comments adequately. I do not have any additional comments on this revision.

7. PLOS authors have the option to publish the peer review history of their article (what does this mean?). If published, this will include your full peer review and any attached files.

Reviewer #1: **Yes: **Jondavid Menteer, MD

Reviewer #2: No

---

## [Editor Report · Acceptance letter]

1 Dec 2022

PONE-D-22-23143R1 

 Abnormalities of hippocampus and frontal lobes in heart failure patients and animal models with cognitive impairment or depression: a systematic review 

Dear Dr. Zhao:

I'm pleased to inform you that your manuscript has been deemed suitable for publication in PLOS ONE. Congratulations! Your manuscript is now with our production department. 

Kind regards, 

on behalf of

Dr. Yoshihiro Fukumoto 

Academic Editor

PLOS ONE